# Evaluation of pregnancy associated glycoproteins assays for on farm determination of pregnancy status in beef cattle

**Adalaide C. Kline[1]☉, Saulo Menegatti Zoca[1‡], Kaitlin M. Epperson[2,3‡], Lacey K. Quail[2,3‡], Jaclyn N. Ketchum[2,3‡], Taylor N. Andrews[4‡], Jerica J. J. Rich[5‡], Jim R. Rhoades[6‡], Julie A. Walker[1‡], George A. Perry[3☉*]**

1 Department of Animal Science, South Dakota State University, Brookings, South Dakota, United States of America, 2 Department of Animal Science, Texas A&M University, College Station, Texas, United States of America, 3 Texas A&M AgriLife Research, Overton, Texas, United States of America, 4 Department of Animal Science, New Mexico State University, Las Cruces, New Mexico, United States of America, 5 Department of Animal Science, Arkansas State University, Jonesboro, Arkansas, United States of America, 6 IDEXX Laboratories, Westbrook, Maine, United States of America

☉ These authors contributed equally to this work.
‡ SMZ, KME, LKQ, JNK, TNA, JJJR, JRR and JAW also contributed equally to this work.
* george.perry@ag.tamu.edu

**Data Availability Statement:** All relevant data are within the manuscript and its Supporting information files.

## Abstract

Transrectal ultrasonography is known as the gold standard for pregnancy detection, but requires costly equipment and technical skills; therefore, access to an inexpensive and more user-friendly method with similar accuracy could benefit cattle producers. Detection of pregnancy-associated glycoproteins can accurately determine pregnancy in ruminants; however, usually requires specialized equipment for the assay. Thus, the objectives of these studies were to 1) validate the IDEXX Alertys OnFarm Pregnancy Test (lateral flow) and compare the accuracy of all three commercial PAG assays to transrectal ultrasonography and 2) to determine the postpartum interval necessary for clearance of pregnancy-associated glycoproteins from the previous pregnancy to avoid false positives. In study 1, blood samples from previously identified pregnant *Bos taurus* females from six different herds (nulliparous n = 1,205 and multiparous n = 1,539; samples collected between d 27 to 285 of gestation over a three-year period) were utilized. In study 2, postpartum females (primiparous n = 48 and multiparous n = 66) from one herd were utilized: (n = 1,066; samples collected weekly for up to 12 weeks postpartum). In study 1, level of agreement between different methods of pregnancy detection was determined by Pearson's correlation and Kappa scores. In study 2, data were analyzed as a repeated measure using the MIXED procedure of SAS with main effects of parity, days postpartum (dpp), and parity by days postpartum, then data were analyzed further using the REG procedure of SAS. In study 1, transrectal ultrasonography and lateral flow were positively correlated (r = 0.77; P <0.01), with 92.4% agreement. In study 2, the abundance of absorbance of PAGs rapidly decreased from 0 to 50 days postpartum, then continued to gradually decrease (P <0.01; r = 0.90). Prior to 42 days postpartum, PAG concentrations were sufficiently elevated resulting in

**Funding:** This project was funded in part by South Dakota Hatch Funds and Texas A&M AgriLife Multistate Hatch project 9835. Products were contributed by IDEXX and Zoetis.

**Competing interests:** I have read the journal's policy and the authors of this manuscript have the following competing interests: Jim Rhoades is employed by IDEXX.

false positive readings in all assays. In conclusion, there is very good agreement between transrectal ultrasonography and PAG assays, but likelihood of false positive results are highif assays are performed fewer than 42 days postpartum.

## Introduction

Pregnancy diagnosis within an operation is not only important, but necessary to increase profitability and have a complete and successful reproductive management program [1]. Fertilization occurs greater than 90% of the time following insemination of beef cows that have been detected in estrus, but calving rates to a single insemination are usually only about 55% [2]. According to recent USDA NAHMS data, only 31.6% of all cow-calf operations use a method of pregnancy determination [3], and given the discrepancy between fertilization rate and calving rate, an accurate pregnancy detection method is critical to maximizing herd profitability and production efficiency. In order to increase the percentage of operations that utilize a pregnancy detection method; however, it must be accurate and easy to use.

Transrectal ultrasonography is considered the gold standard for pregnancy detection, but it is costly and requires a skilled technician [4]. An alternative method to determine pregnancy is by detecting pregnancy-associated glycoproteins (PAGs) in circulation [5–8]. Blood pregnancy tests are also increasing in popularity due to ease of use and the unique feature of not requiring costly equipment or special training. Pregnancy-associated glycoproteins are synthesized by trophoblast giant cells (TGCs) of the trophectoderm in the ruminant placenta [9]. Binucleate giant cells then migrate to fuse with maternal uterine epithelial cells where the granular content within the TGC is released into the maternal circulation. Once the granular content is released to maternal circulation, PAGs can be measured in either milk or blood samples to determine pregnancy status [10–12]. These glycoproteins can be detected in the maternal bloodstream as early d 22 of gestation [13, 14].

Pregnancy-associated glycoproteins concentrations steadily increase in the maternal bloodstream throughout gestation, are elevated at time of parturition, and then decrease after parturition [5, 6, 15]. They also have a long half-life in the blood of postpartum females, ranging from 80 to 100 days postpartum (dpp) [7, 16]. Because PAGs peak at parturition and have a long half-life, residual concentrations can still exist in both primiparous and multiparous animals at the start of the subsequent breeding season. Thus, when trying to use PAG concentrations as a marker for early pregnancy diagnosis, these residual concentrations may impact the result. Therefore, the objectives of these studies were to 1) validate the IDEXX Alertys OnFarm Pregnancy Test (lateral flow) and compare the accuracy of all three commercial PAG assays to transrectal ultrasonography and 2) to determine postpartum interval necessary for clearance of PAGs from the previous pregnancy to avoid false positives.

## Materials and methods

All procedures were approved by the South Dakota State University Institutional Animal Care and Use Committee (IACUC number17-046A, 18-014E and 1910-061E).

### Experimental design

In study 1, blood samples (nulliparous n = 1,205 and multiparous n = 1,539) from six different *Bos taurus* herds in the state of South Dakota were utilized. Blood samples were collected over a three-year period (2018, 2019, and 2020) from d 27 to 285 of gestation. Pregnancy detection

was performed by transrectal ultrasonography between d 30 and 80 post-insemination in all animals.

In study 2, blood samples from Angus and Angus-cross postpartum females (primiparous n = 48 and multiparous n = 66) from one herd in South Dakota were utilized. Blood samples were collected once a week for up to 12 weeks postpartum (n = 1,066 samples; range of first and last sample was 1–7 to 85–91 dpp).

## Blood sampling

**Serum.** Blood samples were collected by venipuncture of either the coccygeal or jugular vein into 10-mL Vacutainer tubes (Becton, Dickinson and Company, Franklin Lakes, NJ) and stored at room temperature (20°C) for approximately 2 h until centrifuged. Samples were centrifuged at 2,000 x g for 30 min, serum was collected and stored at -20°C until PAG assays were conducted.

**Plasma.** Blood samples were by venipuncture of either the coccygeal or jugular vein into 10-mL EDTA Vacutainer tubes containing (Becton, Dickinson and Company, Franklin Lakes, NJ) and were immediately placed on ice. Samples were centrifuged (1200 × g for 20 min at 4 °C) within 2 h of collection, and plasma was aspirated and stored at -20°C until PAG assays were conducted.

## Transrectal ultrasonography

All animals were evaluated for pregnancy by transrectal ultrasonography by a trained technician with an Ibex EVO ultrasound and 5 MHz linear array probe on d 28 following their first insemination. Pregnancy diagnosis was based on the visualization of an embryo or absence of one. A final pregnancy diagnosis occurred between d 30 and 80 following the end of the breeding season to determine if early fetal loss occurred.

## IDEXX Alertys Rapid Visual Pregnancy Test (RVPT)

Pregnancy was determined in whole blood and serum samples utilizing the commercially available blood pregnancy tests, IDEXX Alertys Rapid Visual Pregnancy Test (IDEXX, Westbrook, ME) according to the manufacturer's directions. Briefly, positive/negative controls, and samples were pipetted into coated plates, and plates were washed and treated with reagents according to the manufacturer's instructions. Visual evaluation of the plates based on a numerical scale, established by color intensity were made upon completion of the procedure by one technician. Color intensity evaluation and description were described by Northrop et al. [17]. The scoring system included a yes/no assignment and numerical value from 0–3 based on color intensity in comparison to the positive and negative control wells, where a score of 0 had the same or less color than the negative control, a score of 1 had slightly more color than the negative control, a score of 2 had slightly less color than the positive control, and a score of 3 had the same or more color than the positive control. A numerical score of 0 or 1 would result in "no" meaning the female is not pregnant, while a numerical score of 2 or 3 would result in "yes" meaning the female is pregnant [17]. This scoring system is not provided by the manufacture with the kit, but instead it was an internal lab assessment.

## IDEXX Alertys Ruminant Pregnancy Test (RPT)

Pregnancy was determined in samples using the commercially available blood pregnancy tests, IDEXX Alertys Ruminant Pregnancy Test (RPT; IDEXX, Westbrook, ME) according to the manufacturer's directions. Briefly, controls and serum samples were pipetted in duplicate into

wells of the coated plates, and plates were washed and treated with reagents according to the manufacturer's instructions. The results from the RPT were analyzed on a Molecular Devices SpectraMax 190 microtiter plate reader that measures the optical density of the wells (San Jose, California). The interassay CV was 3.8% and 3.7% for plasma and serum respectively. The intraassay CV was 2.82%, and the cutoff for pregnancy was a S-N value of $\geq 0.300$.

### IDEXX Alertys OnFarm Pregnancy Test (lateral flow)

Pregnancy was determined using commercially available blood pregnancy tests, IDEXX Alertys OnFarm Pregnancy Test (IDEXX, Westbrook, ME) according to the manufacturer's directions. Inside an IDEXX Alertys OnFarm Pregnancy Test kit includes a lateral flow test, pipette, and a dropper with chase buffer. Briefly, 150 μL of serum or plasma was pipetted into the well of the lateral flow test followed with 175 μL of chase buffer. After a 20 min incubation, the tests were scored and evaluated by two technicians that were blind to ultrasonography pregnancy status. A third technician was utilized to break any discrepencies. Interpretation of IDEXX Alertys OnFarm Pregnancy Test with just the internal positive control, "C", visible line indicates the female was not pregnant, while visibility of the test sample, "T", line means the female was pregnant at time of blood sample (S1 Fig).

### Statistical analysis

In study 1, blood samples were analyzed to determine pregnancy status agreement between tests (Ultrasonography, RVPT, RPT and lateral flow) using the CORR procedure of SAS (9.4). Since the correlation between tests was significant, further analysis were performed using the FREQ procedure of SAS to evaluate the frequency of pregnant and open between each test comparatively to each other. Further analysis include measurements of specificity, sensitivity, positive prediciticve value, negative predictive value, and percent correct between the pregnancy detection methods with herd as the random effect. The Kappa scoring scale utilized is as follows: 0.80–1.00 = Very good, 0.60–0.80 = Good, 0.40–0.60 = Moderate, 0.20–0.40 = Fair, and < 0.20 = Poor. Statistical significance was considered at $P \leq 0.05$, and a tendency at $0.05 < P \leq 0.10$.

In study 2, samples were grouped by postpartum week of collection. Pregnancy status/optical density readings were analyzed as a repeated measure using the MIXED procedure of SAS (9.4) using the compound semetry covariant structure. The statistical model consisted of time postpartum, parity, and their interactions. The effect of parity was analyzed using animal within treatment as the error term, and effects of time and the interaction were analyzed using the residual as the error term. The effect of dpp on PAG concentrations (pregnancy status/ optical density) using the REG procedure of SAS. Statistical significance was determined at $P \leq 0.05$, and a tendency at $0.05 < P \leq 0.10$. All data are reported as (LSmeans ± SEM).

## Results

### Study 1

Agreement based on Kappa scores was very good among all tests in the study (Table 1). Additionally, there was a positive correlation ($P < 0.05$) among all tests (Ultrasonography:RVPT $r^2 = 0.81311$, Ultrasonography:RPT $r^2 = 0.83856$, Ultrasonography:lateral flow $r^2 = 0.84126$, RVPT:RPT $r^2 = 0.94739$, RVPT: lateral flow $r^2 = 0.90138$, RPT:lateral flow $r^2 = 0.91257$; Table 2). Of the 1,096 animals that were diagnosed nonpregnant by transrectal ultrasonography, less than 7% were diagnosed pregnant by any of the PAG assays. Of the 1,648 animals diagnosed pregnant by transrectal ultrasonography, less than 3% were diagnosed nonpregnant

**Table 1. Agreement between pregnancy tests to determine accuracy of pregnancy detection.**

| Test | Ultrasonography[1] | RVPT[2] | RPT[3] | Lateral Flow[4] |
|---|---|---|---|---|
| **Ultrasonography[1]** | | 0.8108 | 0.8344 | 0.8388 |
| **RVPT[2]** | very good | | 0.9472 | 0.9005 |
| **RPT[3]** | very good | very good | | 0.9122 |
| **Lateral Flow[4]** | very good | very good | very good | |

Values depicted above the diagonal line are the Kappa scores, while values below the diagonal are the overall agreement of the tests based on the Kappa score. The Kappa scoring scale is: 0.80–1.00 = Very good, 0.60–0.80 = Good, 0.40–0.60 = Moderate, 0.20–0.40 = Fair, and < 0.20 = Poor.

[1] Ultrasonography = transrectal ultrasonography

[2] RVPT = IDEXX Alertys Rapid Visual Test

[3] RPT = IDEXX Alertys Ruminant Pregnancy Test

[4] Lateral Flow = IDEXX Alertys OnFarm Pregnancy Test

**Table 2. Correlation between pregnancy tests to determine accuracy of pregnancy detection.**

| Test | Ultrasonography[1] | RVPT[2] | RPT[3] | Lateral Flow[4] |
|---|---|---|---|---|
| **Ultrasonography[1]** | | 0.81311 | 0.83856 | 0.84126 |
| **RVPT[2]** | < 0.0001 | | 0.94739 | 0.90138 |
| **RPT[3]** | < 0.0001 | < 0.0001 | | 0.91257 |
| **Lateral Flow[4]** | < 0.0001 | < 0.0001 | < 0.0001 | |

Values depicted above the diagonal line are the correlation coefficients, $r^2$, of the tests, while values below the diagonal are the *P*-values of all the tests. A positive correlation and significant difference were found among the tests in comparison to each other.

[1] Ultrasonography = transrectal ultrasonography

[2] RVPT = IDEXX Alertys Rapid Visual Test

[3] RPT = IDEXX Alertys Ruminant Pregnancy Test

[4] Lateral Flow = IDEXX Alertys OnFarm Pregnancy Test

by any of the PAG assays. Thus, a greater than 90% agreement occurred between transrectal ultrasonography and all of the PAG assays (Table 3). Comparisons were also made between the three PAG assays, and a greater than 95% agreement occurred between all assays (Table 3).

## Study 2

**PAG clearance.** When using the RPT assay, there were no differences detected in PAG concentrations between primiparous and multiparous females (parity; $P = 0.73$) and parity by dpp ($P = 0.55$); however, there was a significant effect of dpp group on PAG concentrations in postpartum females ($P < 0.01$). A linear decrease from 0 to 50 dpp occurred, and then PAG concentrations reached a sustained nadir from 50 to 84 dpp. There was a strong correlation between dpp group and PAG concentrations, accounting for 67.48% of the variance. Since PAGs reached a sustained nadir after 50 dpp, statistical analysis was performed to determine the clearance from 0 (calving) to 50 dpp. Elimination of the samples when they first reached a sustained nadir allowed for a stronger correlation between PAG concentrations and dpp, accounting for 80.83% of the variance ($P < 0.01$). When determining average PAG concentrations of samples by dpp group broken down into 7 d intervals through d 84, clearance below pregnancy threshold of the RPT assay occurred by 42 dpp (optical density (od) = $0.26 \pm 0.036$, Fig 1A). Animals were considered pregnant when od were $\geq 0.3$. There was no significant

**Table 3. Agreement between pregnancy detection methods to determine accuracy among methods.**

| Test[1] | Agreement, % | False Positive[2], % | False Negative[3], % | Sensitivity, %[4] | Specificity, %[5] | Samples, n |
|---|---|---|---|---|---|---|
| Ultrasonography[6]:Lateral Flow[7] | 92.38 | 5.61 | 2.00 | 97.2 | 85.9 | 2,744 |
| Ultrasonography[6]:RVPT[8] | 90.73 | 6.46 | 2.80 | 93.1 | 89.1 | 1,533 |
| Ultraound[6]:RPT[9] | 92.61 | 5.91 | 1.48 | 97.7 | 83.4 | 2,436 |
| RVPT[8]:Lateral Flow[7] | 95.07 | 3.56 | 1.37 | 96.9 | 92.9 | 1,460 |
| RPT[9]:Lateral Flow[7] | 96.22 | 1.31 | 2.46 | 96.5 | 95.7 | 2,360 |
| RPVT[8]:RPT[9] | 97.36 | 1.80 | 0.83 | 98.2 | 96.6 | 1,443 |

[1]Comparison between tests first:second

[2] False Positive = a result that shows a female is pregnant when she is actually non-pregnant

[3]False Negative = a result that shows a female is non-pregnant when she is actually pregnant

[4]Sensitivity = number of true positives divided by number of true positives plus number of false negatives

[5]Specificity = number of true negatives divided by number of true negatives plus number of false positives

Ultrasonography = transrectal ultrasonography

[5]Lateral Flow = IDEXX Alertys OnFarm Pregnancy Test

[6]RVPT = IDEXX Alertys Rapid Visual Test

[7]RPT = IDEXX Alertys Ruminant Pregnancy Test

difference in determining the clearance of PAGs with the IDEXX Alertys Ruminant Pregnancy Test between both multiparous and primiparous females ($P = 0.55$; Fig 1B).

For postpartum samples analyized by the RVPT, there was a significant difference ($P = 0.04$) in PAGs between parity, an effect of dpp ($P < 0.01$), and a dpp by parity interaction ($P = 0.03$). All animals regardless of parity were still considered pregnant from the previous pregnancy through 21 dpp (98.63 ± 2.62%), whereas by 28 dpp, 88.36 ± 2.58% were considered pregnant (Fig 2A). The percentage of females that received a false positive pregnancy diagnosis rapidly decreased as dpp increased. By 49 dpp, 11.82 ± 2.64% of the females were considered positive for pregnancy, and at 56 dpp, there were 1.98 ± 2.70% positive for pregnancy (Fig 2A). The detection of false positives rapidly decreased from 21 to 56 dpp, then sustained nadir from 56 to 84 dpp (Fig 2A). There was a significant difference in the clearance of PAGs considering the parity by dpp interaction ($P = 0.03$; Fig 2B). Between 35 to 49 dpp, there was a greater decrease in false positives among primiparous compared to multiparous animals (at 49 dpp 5.01 ± 4.02% and 18.63 ± 3.43%; respectively). At 84 dpp 7.85 ± 6.52% of primiparous and 1.77 ± 6.14% of multiparous females were still considered pregnant (Fig 2).

For postpartum samples analyzed by the lateral flow test, there was no difference ($P = 0.21$) of parity, but there was an effect of dpp ($P < 0.01$) and a tendency for a dpp by parity interaction ($P = 0.06$). All animals regardless of parity were still considered pregnant from the previous pregnancy through 35 dpp (100 ± 2.58%), whereas by 42 dpp, 98.16 ± 2.55% were considered pregnant (Fig 3A). The percentage of females that received a false positive pregnancy diagnosis declined as dpp increased. By 77 dpp, there were 13.72 ± 3.16% of the females positive for pregnancy, and at 84 dpp, there were 4.11 ± 4.39% positive for pregnancy (Fig 3A). The detection of false positives rapidly decreased between 42 and 70 dpp group, then slowly decreased from 70 to 84 dpp group (Fig 3A). There was a tendency for a parity by dpp interaction ($P = 0.06$; Fig 3B). Between 63 to 77 dpp group there was a greater decrease in false positives among primiparous compared to multiparous animals (at 77 dpp 5.12 ± 4.26% and 22.31 ± 4.67%; respectively). At 84 dpp 3.56 ± 6.38% of primiparous and 4.66 ± 6.03% of multiparous females were still considered pregnant (Fig 3).

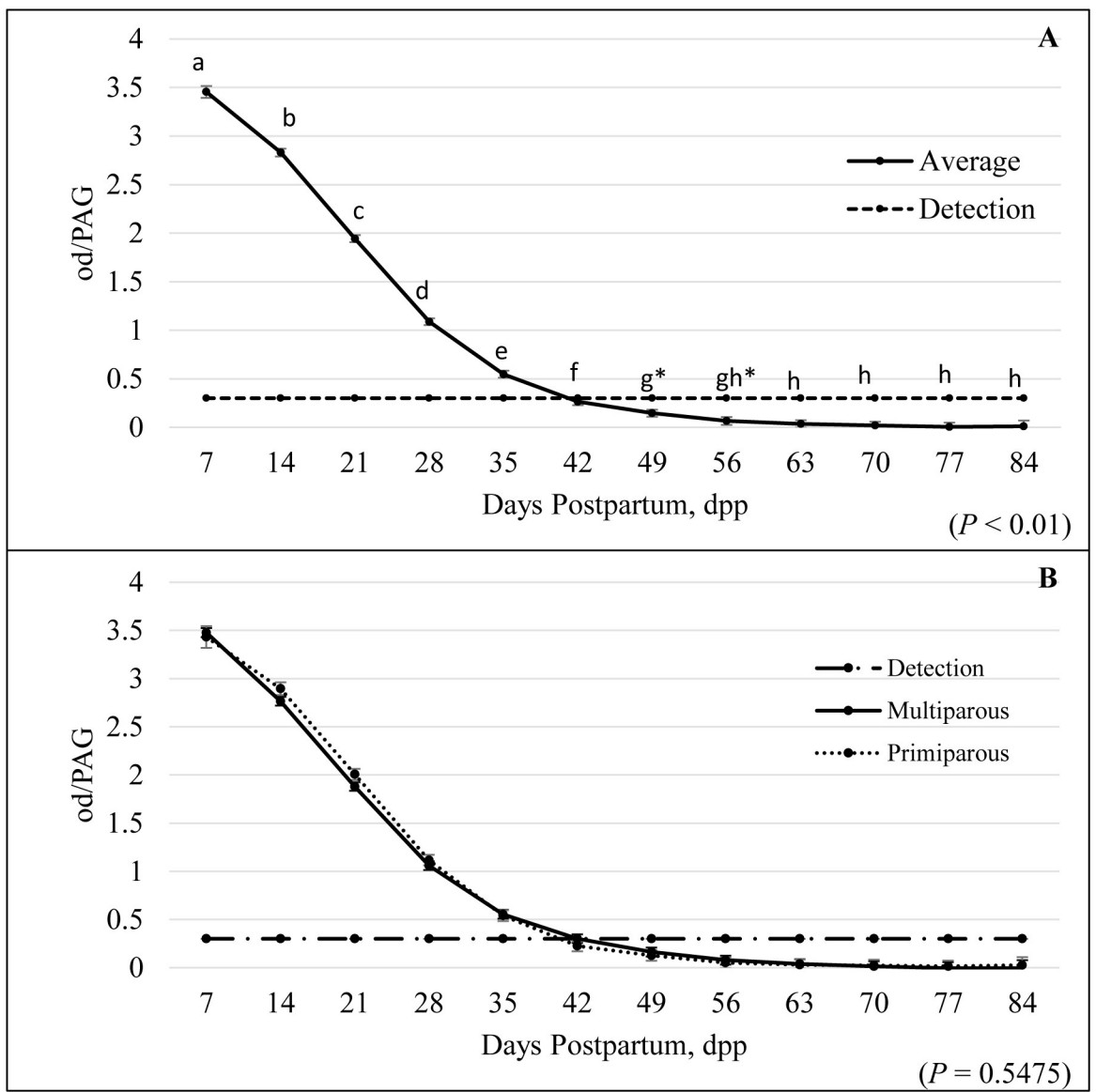

**Fig 1. Clearance of PAG concentrations in postpartum beef cattle (A) and overall comparison between parity (B).** Mean (± SEM) serum pregnancy-associated glycoprotein (PAG) concentration levels among postpartum beef females (A) for the Ruminant Pregnancy Test. PAG concentration levels were below the detectable range by 42 dpp (optical density (od) = 0.2636; A). Values not sharing the same superscripts (a-h) differ ($P < 0.01$; A). Superscript (*) represents values that tended to differ ($P = 0.08$; A). There was no significant difference in clearance patterns among postpartum multiparous and primiparous beef females ($P = 0.55$; B).

## Discussion

Currently, there is no pregnancy detection method that is 100% accurate for pregnancy determination without being invasive (i.e. harvesting the reproductive tract), which makes it difficult to evaluate the accuracy of a new pregnancy detection method [18]. The most accurate, minimally-invasive (i.e. without harvesting the reproductive tract) method currently known as the gold standard, is transrectal ultrasonography. Efforts have been ongoing to develop a method which accurately determines pregnancy status and is also producer friendly and quick to use. IDEXX Laboratories have created a series of blood pregnancy tests, (Alertys Rapid

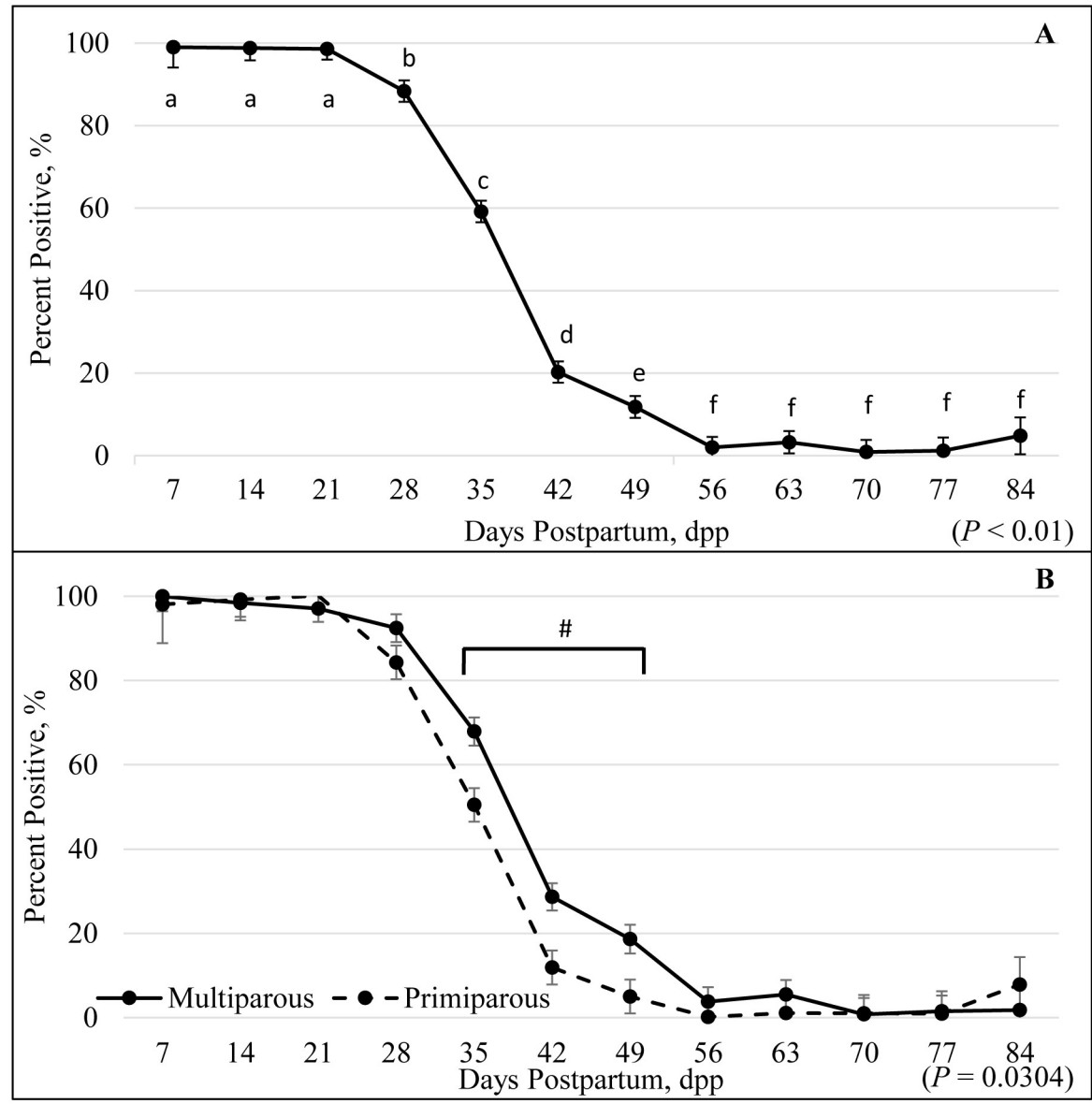

**Fig 2. False-positive percentage by days postpartum (A) and parity (B) utilizing the Rapid Visual Pregnancy Test.** Postpartum samples were analyzed by the Rapid Visual Pregnancy Test (RVPT) to determine an accurate timeframe to test pregnant females without getting a false positive test from the residual pregnancy-associated glycoproteins (PAGs). There was a significant effect of days postpartum (dpp; $P < 0.01$; A). All animals were still considered pregnant from the previous pregnancy on 21 dpp (98.63%; A). There was a significant difference of dpp by parity ($P = 0.03$; B). # values between multiparous and primiparous between 35 to 49 dpp differed $P < 0.05$.

Visual; Alertys Ruminant Pregnancy Test; and Alertys OnFarm Pregnancy Test), to help producers determine pregnancy status of females within their herd. The RVPT and RPT utilize polyclonal antibodies, allowing for the detection of various members of the PAG family that are secreted, some at different times throughout gestation, and found in the maternal bloodstream to determine pregnancy status. The lateral flow test utilizes a monoclonal antibody, meaning the antibody used binds to specific members of the PAGs family (proprietary information) that is secreted in order to determine the pregnancy status of the female.

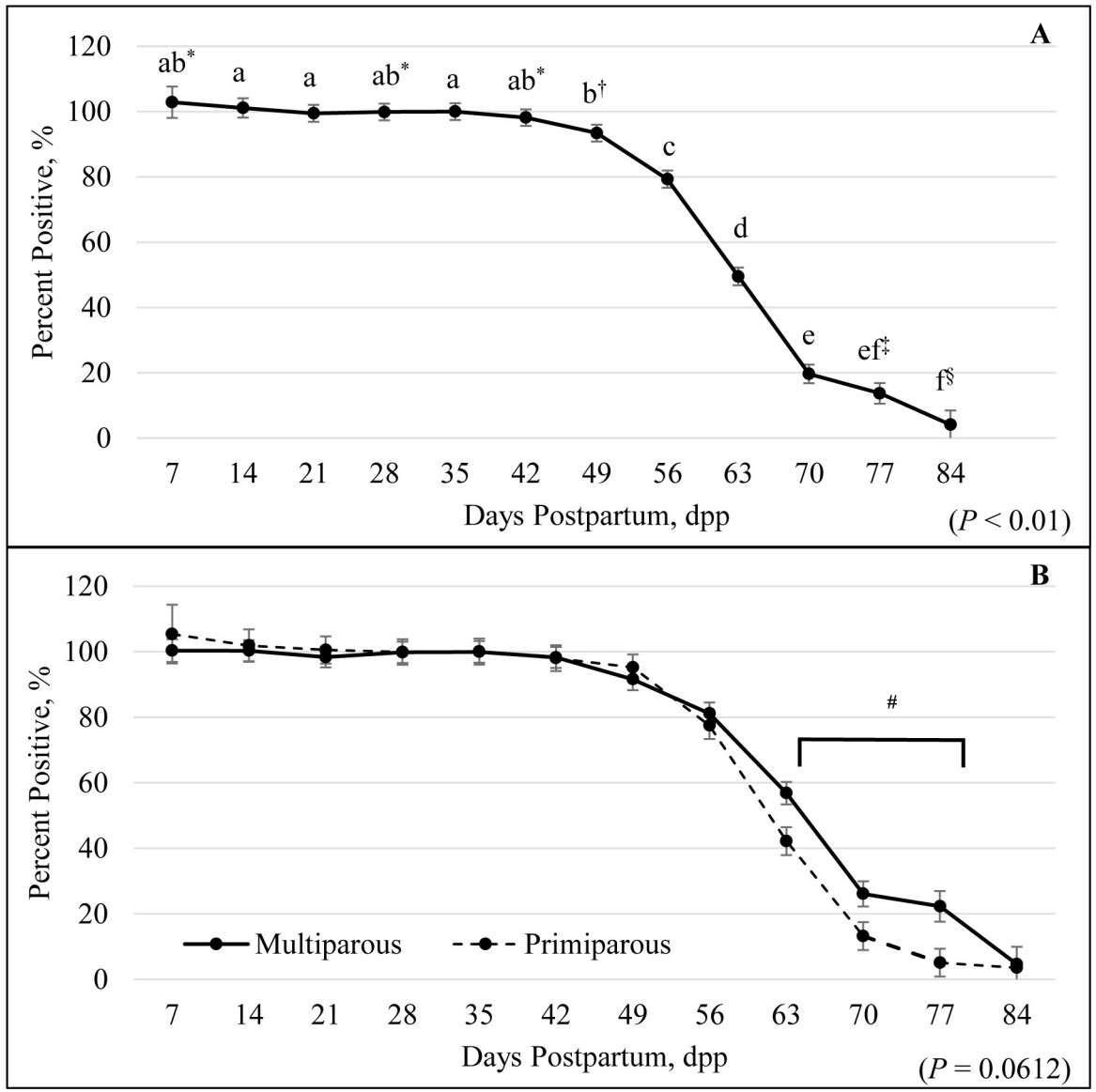

**Fig 3. False positive percentage by days postpartum (A) and parity (B) utilizing the OnFarm Pregnancy Test.** Postpartum samples were analyzed by the OnFarm Pregnancy Test (lateral flow) to determine an accurate timeframe to test pregnant females without getting a false positive test from the residual pregnancy-associated glycoproteins (PAGs) from the previous pregnancy. There was a significant effect of dpp ($P < 0.01$; A). All animals were still considered pregnant from the previous pregnancy on 35 dpp (100%; A). Days postpartum by parity tended to be different ($P = 0.06$; B). a-f values not sharing the same superscripts differed $P < 0.05$. *† values not sharing the same superscripts tended to differ $P \leq 0.08$. ‡§ values not sharing the same superscripts tended to differ $P \leq 0.07$. # values between multiparous and primiparous between 63 to 77 dpp differed $P < 0.05$.

In order for a pregnancy detection test to be beneficial, it must be sensitive, meaning it is accurate in identifying pregnant females, and specific, meaning it also accurately identifies females who are nonpregnant. Transrectal ultrasonography was used as a comparative measure to determine the accuracy of all the IDEXX PAG pregnancy tests as it is considered the gold standard.

Epperson et al. [19] conducted a comparison between RVPT, ultrasonography, RPT and resynchronization pregnancy diagnosis. Blood samples were taken on d 28 and the final

pregnancy diagnosis was made 31 to 80 d post-second artificial insemination (AI2). The kappa statistic scores among the comparisons between RVPT, ultrasonography, and RPT against the final resynchronization pregnancy diagnosis were very good; 0.90, 0.82, and 0.90, respectively. In a study completed by Silva et al. [20], the accuracy of PAG ELISA to transrectal ultrasonography on d 27 post-AI, d 39 post-AI2, and d 39 post-AI3 (post-third artificial insemination) was compared utilizing dairy cows. These comparisons found the PAG blood pregnancy test to have a kappa statistic score of 0.87 to 0.90, which is very good, when compared to ultrasonography. Similarly, a study by Romano and Larson [18] compared pregnancy specific protein B (PSPB; PAG-1 subgroup) ELISA to transrectal ultrasonography on d 28, 30, and 35 post-AI. Romano and Larson [18] found between d 28 to d 35 PSPB compared to transrectal ultrasonography had a very good kappa statistic score of 0.93 for accurately detecting pregnancy status. Piechotta et al. [21] utilizing dairy cows, compared two ELISA blood pregnancy tests for PSPB) against transrectal ultrasonography between d 26 to 58 post-AI. Between the two tests there was no significant difference found in comparison to transrectal ultrasonography. Northrop et al. [17] also reported similar results when comparing the RPT and RVPT to transrectal ultrasonography in beef cows between d 28 to 40 post-AI, and found them to have very good agreement, 0.86 and 0.85 respectively. The current study found similar results when comparing RVPT and RPT to ultrasonography with very good kappa scores (0.81 and 0.83, respectively) and agreement (81.3% and 83.9%, respectively). Similarly, validating the IDEXX Alertys OnFarm Pregnancy Test to ultrasonography, RVPT, and RPT, there was very good kappa scores (0.84, 0.90, 0.91, respectively) and agreement (84.1%, 90.1%, and 91.3%, respectively). The results from the current study along with previous research highlight that blood pregnancy tests are highly accurate compared to transrectal ultrasonography. Considering the lateral flow test had a very good kappa score and agreement among all three tests used in the present study, the lateral flow test would make a great pregnancy detection alternative to the costly ultrasound equipment or laboratory equipment to run the RPT. The costs associated with the blood test methods range from $4.50 to $8 per head while the cost for ultrasound is going to depend on the veterinarian's rate to perform the test and their speed since many veterinarians charge an hourly rate. A skilled ultrasound technician can also provide information about the pregnancy (viability, stage of gestation, fetal sex, etc) that is not provided by a blood test. In areas where skilled ultrasound technitions are not readally available; however, or when animals numbers do not make them cost effective the use of a cow side test will allow producers to benefit from pregnancy diagnosis and identification of non-pregnant cows which will help their bottom line.

Caution should be used when implementing the lateral flow test to minimize false positives from performing the test too early in gestation, unlike the RVPT or RPT. It would be advised to perform the lateral flow test on d 40 of gestation or later to receive the most accurate results, thus, not keeping or rebreeding a female in the herd who happens to be not pregnant. Commercial PAG tests have a 1 to 5% false positive rate compared to transrectal ultrasonography [22]. These false positives could be due to residual PAG concentrations from the previous pregnancy or from the current embryo being lost.

Due to residual concentrations of PAGs after parturition, blood pregnancy tests need to be conducted at the appropriate time postpartum to accurately determine pregnancy status and avoid false positives. Previous research suggested that PAG pregnancy blood tests should not be utilized until 91 to 120 dpp [7, 8]. Specifically, Zoli et al. [7] used $boPAG_{67\ kDa}$, which is part of the boPAG-1 group that is detectable in the maternal bloodstream through 100 dpp [10]. Others found that PAG pregnancy tests should not be performed until 70 to 90 dpp or later [16, 23, 24]. However, after 80 dpp, Kiracofe et al. [16], determined PAG concentrations were <1 ng/mL, which indicates PAGs from the previous pregnancy would not produce a false

positive result. The differences reported between authors could be due to the different PAGs that were detected in each study.

The RPT resulted in fewer false positives at 42 dpp, thus allowing for earlier testing in gestation compared to the RVPT and lateral flow test evaluated in our laboratory. The utilization of the RPT can be implemented as early as d 28 of gestation, so the RPT could be conducted at 58 dpp or 28 d post-AI to allow for uterine involution to occur [25] and improve accuracy of results. Clearance of residual PAGs from the previous pregnancy determined by the RVPT test took longer than the clearance within the RPT. Also, the clearance of residual PAGs in the lateral flow test surpassed the length of time needed for uterine involution to occur (30 dpp) as well as the manufacturer's recommended utilization day, d 28 post-AI. For the most accurate results (less than 20% false positive) an allotment of 49 dpp or 19 d post-AI and 70 dpp or 40 d post-AI should be made before performing the RVPT and lateral flow test, respectively. Between the RPT and lateral flow, to decrease the amount of false pregnancy diagnoses there is at least a 12 dpp difference if the RPT is utilized on d 28 and the lateral flow is utilized on d 40 of gestation. The RPT uses a polyclonal antibody against several PAGs (e.g. PAGs 4, 6, 9, 16, 18, 19; described in US Patent no. 7,604,950B2; [26]). Specifically, PAGs 4, 9, and 6 are secreted from d 25 (PAG 4 and 9) and d 45 (PAG 6) through d 250 of gestation [27]. The lateral flow test is a monoclonal antibody test where the PAGs identified is not stated (proprietary information). Thus, the difference in detection of PAGs between the two tests may be due to the influence in clearance of different PAGs from circulation.

Performing the lateral flow later in gestation when there would be fewer false positives would effectively detect embryonic loss that occurs through d 45 [28], and may improve management decisions and reproductive opportunities. Implementation of the lateral flow test on postpartum females, may potentially increase the calving interval compared to the RPT test since pregnancy determination would occur later in gestation, causing the possibility of rebreeding, if utilizing AI, to be later as well. In nulliparous females, implementation of the lateral flow test can occur on d 28 of gestation as the manufacturer's instructions state, like the two other tests (RPT and RVPT), since there are no residual PAGs from a previous pregnancy.

For practical and research use in the cattle industry, any female that is diagnosed by a pregnancy detection method as nonpregnant, the female may receive an injection of prostaglandin-$F_{2\alpha}$. Receiving an injection of prostaglandin-$F_{2\alpha}$ would cause luteal regression decreasing progesterone concentrations and allow for the dominant follicle to increase estradiol production and estrus to be initiated [29]. Females falsely classified as nonpregnant by a pregnancy detection method may potentially receive a treatment with prostaglandin-$F_{2\alpha}$, which could cause abortion. Therefore, producers with unintentionally aborted females due to pregnancy misdiagnosis would experience experience an economic loss of $550 to $800 for the loss of a live calf, resynchronization drugs, time, and labor [18, 30]. Silva et al. [20] found the RPT to have a great false negative predicted value from the three different AI days, d 27 post-AI, d 39 post-AI2, and d 39 post-AI3 (97.1%, 96.9%, 97.7% respectively). This study also found a significant negative predictive value between ultrasonography:lateral flow, ultrasonography:RVPT, and ultrasonography:RPT (98.0%, 97.2%, and 98.5%, respectively). Having a greater negative predictive value decreases the likelihood of giving prostaglandin-$F_{2\alpha}$ to a female who is truly pregnant. On the other hand, early and accurate pregnancy detection allows for more management decisions to be made by identifying females who are nonpregnant post-breeding. Animals that are accurately detected as nonpregnant can be rebred or culled earlier to reduce the number of days a nonpregnant female is cared for which ultimately would result in maximizing an operation's economic gains and minimize reproduction associated financial losses [31].

## Conclusion

The utilization of the IDEXX Alertys OnFarm Pregnancy Test (lateral flow) is a competitive alternative in pregnancy detection compared to the gold standard transrectal ultrasonography with an 92.38% agreement comparison in postpartum females. Of the three IDEXX Laboratories tests available, the lateral flow test is the most user-friendly method. Due to the additional time required for diagnosis of pregnancy with blood based tests; however, management decisions may be delayed compared to ultrasonography and may result in additional labor to sort open females once test results are available.

Concentrations of PAGs decreased rapidly for the first 3 weeks after parturition, and after 42 dpp PAG concentrations fell below the concentrations for pregnancy detection using the RPT assay. Thus, there is more confidence gained by the results received from a PAG blood pregnancy test when it is performed at least 42 dpp. More caution should be used with the utilization of the RVPT and lateral flow test postpartum in primiparous and multiparous females until at least 49 or 70 dpp, respectively, due to residual PAGs. Based on the present study, to decrease the likelihood of false positives, pregnancy detection by the lateral flow test should be preformed at d 40 of gestation on beef cattle who were bred at or after 30 dpp, which is later than the mansufacture's recommended day, d 28. Based on results found with the current study utilizing the RVPT and RPT for pregnancy detection, there can be confidence gained on the manufacture's recommended day of use, d 28, with decreased false positive results.

## Supporting information

**S1 Fig. Comparison of a pregnant and not pregnant IDEXX Alertys OnFarm Pregnancy Test, lateral flow.** The test on the left indicates that particular female at the timepoint the test was taken is pregnant due to the visibility of the test sample, "T", line. The test on the right indicates that particular female is not pregnant due to there only being one line visible and that is the internal positive control, "C", line. If the C line does not show up at all the test is invalid and the sample should be reran on a different test.
(DOCX)

**S1 File. Data file 1.** This is the S1 data file.
(DOCX)

**S2 File. Data file 2.** This is the S2 data file.
(DOCX)

## Acknowledgments

The authors would like to thank IDEXX Laboratories for the donation of the IDEXX Alertys Ruminant Pregnancy Test and IDEXX Alertys OnFarm Pregnancy Test, as well as the cooperator herds for the use of their cattle, and zoetis for synchronization products.

## Author Contributions

**Conceptualization:** Adalaide C. Kline, George A. Perry.

**Data curation:** Adalaide C. Kline, Saulo Menegatti Zoca, George A. Perry.

**Formal analysis:** Adalaide C. Kline, George A. Perry.

**Funding acquisition:** Jim R. Rhoades, Julie A. Walker, George A. Perry.

**Investigation:** Adalaide C. Kline, Saulo Menegatti Zoca, Kaitlin M. Epperson, Lacey K. Quail, Jaclyn N. Ketchum, Taylor N. Andrews, Jerica J. J. Rich, George A. Perry.

**Methodology:** Adalaide C. Kline, George A. Perry.

**Project administration:** George A. Perry.

**Resources:** Jim R. Rhoades, Julie A. Walker, George A. Perry.

**Software:** George A. Perry.

**Supervision:** George A. Perry.

**Validation:** Adalaide C. Kline, George A. Perry.

**Visualization:** Adalaide C. Kline, Saulo Menegatti Zoca, George A. Perry.

**Writing – original draft:** Adalaide C. Kline, Saulo Menegatti Zoca, George A. Perry.

**Writing – review & editing:** Adalaide C. Kline, Saulo Menegatti Zoca, Kaitlin M. Epperson, Lacey K. Quail, Jaclyn N. Ketchum, Taylor N. Andrews, Jerica J. J. Rich, Jim R. Rhoades, Julie A. Walker, George A. Perry.

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
