## [Decision Letter · Decision Letter 0]

18 Feb 2024

PONE-D-24-00715Evaluation of pregnancy associated glycoproteins assays for on farm determination of pregnancy status in beef cattlePLOS ONE

Dear Dr. Perry,

Thank you for submitting your manuscript to PLOS ONE. After careful consideration, we feel that it has merit but does not fully meet PLOS ONE’s publication criteria as it currently stands. Therefore, we invite you to submit a revised version of the manuscript that addresses the points raised during the review process. Two experts have reviewed your manuscript. They both have raised some moderate to major concerns. I concur with their view and invite you to address all their concerns by revising your manuscript accordingly. Please note that reviewer #1 has appended their comments in an attached file. The file should be attached to this email but can also be accessed through the editorial management portal.

We look forward to receiving your revised manuscript.

Kind regards,

Angel Abuelo, DVM, MRes, MSc, PhD, DABVP (Dairy), DECBHM

Academic Editor

PLOS ONE

Reviewers' comments:

Reviewer's Responses to Questions

**Comments to the Author**

1. Is the manuscript technically sound, and do the data support the conclusions?

Reviewer #1: Yes

Reviewer #2: Yes

2. Has the statistical analysis been performed appropriately and rigorously? 

Reviewer #1: Yes

Reviewer #2: I Don't Know

3. Have the authors made all data underlying the findings in their manuscript fully available?

Reviewer #1: Yes

Reviewer #2: Yes

4. Is the manuscript presented in an intelligible fashion and written in standard English?

Reviewer #1: Yes

Reviewer #2: Yes

5. Review Comments to the Author

Reviewer #1: Overall, the manuscript is straightforward and easy to read. It provides valuable insight on a newly available method of pregnancy diagnosis in ruminants. Additional comments and suggestions are in the attached document.

Reviewer #2: Specific Comments:

Title and Abstract

Appropriate title.

In abstract should probably be inexpensive rather than cheap? Line 27

.. likelihood of false positive results are high if assays are performed fewer than 42 days pp. Might be less confusing (Lines 49-50.)

Introduction

Line 72; only 22days necessary when describing as early as…

Line 73: probably best to describe where the PAG’s are increasing- circulation?

Aims of study clearly stated.

M&M’s

IACUC stated- ethical research.

Study 1

It would be useful to know if those that were diagnosed pregnant via ultrasonography also calved, illustrating no pregnancy loss occurred to validate the PAG assay results. Otherwise the results would need to be assessed for the possibility of pregnancy loss and inherent PAG decreases.

Study 2

Good array of time points.

Ultrasonography

Please state transducer frequency. Line 112

Please also state either conceptus was detected; or an embryo or fetus were detected. The embryo becomes a fetus at between 40-45 days depending on publications and interpretations. Similarly embryonic loss can only occur up to day 45; maybe best to describe as late embryonic or early fetal loss occurred? Lines 114-115.

Lateral Flow

Please update the Figure descriptions, in line 127 the relevant figure is not numbered, and might not be featured in the SI?

RVPT

The scoring for the RVPT appears to be a bit confusing, and should probably be a discussion point?

Statistical analysis

Please use ultrasonography rather than ultrasound. It might be better to compare the tests in the same order as they are introduced: ultrasonography, lateral flow, RVPT and RPT, to make it easier for the reader.

Line 166

My understanding is the ‘tendencies’ should not be part of statistical analysis.

Results

Table 1 useful.

For table 2 and for all others please alter ultrasound for ultrasonography.

Please ensure all the figures are correctly documented in the article, and correlate with the relevant illustrated figures.

Line 282; not sure that tendency should be recorded; maybe in the discussion. Similarly for line 290, and for all mentions of a tendency.

Discussion and conclusion

Line 323; remove ‘a’ or member to singular.

Lines 357-359: It is difficult to interpret the data without knowing the costs of the tests relative to the costs of ultrasonography. And, recognising the skill required to take the correct blood samples. If this could be addressed please.

Lines 360-364:

Please discuss the relevance of a false positive compared with a false negative in relation to the managers requirements- feeding of a non-pregnant animal, or culling of a pregnant animal.

Discussion of relevance of DPP PAG results in relation to the likelihood of pregnancy if insemination is unlikely? And, importance of multiparity cf primiparity esp 35-49d pp. (Why?) (Lines 376-390)

Line 388: Probably does not need: ‘with that said’; and to explain the differences RPT and Lateral Flow options more succinctly.

Line 408: A ‘shot’ although a common colloquialism, should probably be best described scientifically as some sort of drug administration/injection?

I think there needs to be some discussion as to the ‘immediacy’ of the gold standard ultrasonography compared with the relative delays associated with the tests described in the article. Particularly in relation to the handling and management of the animals in relation to the animals that are not pregnant.

Some mention of the likelihood of a bovid in most management systems being inseminated within 50 days, hence the likelihood of being pregnant should be discussed in relation to the levels of PAG in post-partum cows. This in relation to the points mentioned in lines 442-446. (Uterine involution and voluntary waiting periods are typically more than 40 days in most dairy animals, and the effects of the described lactational anoestrus in beef animals typically resulting in breeding not occurring prior to 50days PP.)

Competing Interests

I think it is difficult to believe that no competing interests were declared when Idexx and Zoetis supported the study to a degree, and at least one author is an employee of Idexx.

The Funding declaration in lines 452-453 appears to be at odds with that in the Financial disclosures section of the pre-manuscript section.

Recommendation:

This is useful information that should be published, with minor revisions suggested.

6. PLOS authors have the option to publish the peer review history of their article (what does this mean?). If published, this will include your full peer review and any attached files.

Reviewer #1: No

Reviewer #2: No

---

## [Author Response · Author response to Decision Letter 0]

29 Apr 2024

Dr. Abuelo,

We greatly appreciate the effort you and the reviewers have made in reviewing this manuscript. We have incorporated almost all of the suggested comments, and our individual responses to each suggestion is listed below.

Reviewer #1: Overall, the manuscript is straightforward and easy to read. It provides valuable insight on a newly available method of pregnancy diagnosis in ruminants. Additional comments and suggestions are in the attached document.

Overall, the manuscript is well written and easy to read. It details an evaluation of a newly marketed cow-side pregnancy test. Evaluation of such new tools is important for the beef and dairy industry including veterinarians and producers. Understanding test function, limitations, and accuracy is critically important for those making decisions on whether to use such tools for their herd. There is certainly value in reporting the results of this study. Below are some items to consider to strengthen the manuscript as currently written.

Lines 59-61 – Beef NAHMS 2017 survey from USDA has data on the percentage of beef herds that use ultrasound for pregnancy diagnosis or perform any pregnancy diagnosis at all. According to this data set, only 31.6% of beef operations surveyed used either blood based, ultrasound, or palpation pregnancy diagnosis. Including this data could strengthen your argument for use of a rapid, cow-side test for pregnancy and the impacts on production efficiency and profitability. 

Reference to the NAHMS data has been included.

Line 71 – Roberts et al., Reprod Domest Anim 2015; 50(4):651-8 investigated the use of a milk-based ELISA in beef cows. This could be included here to indicate it’s not only been investigated in dairy cows. 

This reference has been added. 

The ultimate gold standard for comparison of any pregnancy diagnosis method is confirmed delivery of a fetus. Since this study spanned 3 years, do you have calving data for which you would be able to compare and calculate sensitivity and specificity of the pregnancy diagnosis method used relative to calving? This would also allow for some discussion of rates of loss of pregnancy in beef cows which would be valuable to practitioners and producers. 

The authors do have this data on a subset of animals; however, loss among this group was only so low that that analysis of this data would not be valid. 

Reporting the loss rates between the 28d ultrasound examination and the recheck, as well as how this compared to the results of the assays would be useful and practical information for users of these tests.

Only 68 lost a pregnancy between the first and final pregnancy diagnosis. With these low numbers it is not possible to evaluate the data as a predictive test. When a retrospective analysis is done values will appear different, but as a predictive value there is too much variation in PAG concentrations to predict which animals might lose a pregnancy.

Lines 145-147 – We intra- and inter-assay coefficients of variation calculated for the OD readings on the RPT? What were the cutoffs used for pregnant, non-pregnant, and inconclusive/recheck?

The inter and intra assay CVs have been included “The interassay CV was 3.8% and 3.7% for plasma and serum respectively. The intraassay CV was 2.82%, and the cutoff for pregnancy was a S-N value of ≥0.300.” 

There are no sensitivities or specificities calculated in this study. You have only reported false negative and false positive results with NPV and PPV which can be impacted by the population of this one study (ie prevalence of open or pregnant cows in your study population). The sensitivity and specificity, along with confidence intervals, should be included, as this reflects the expected performance of the test. Previous studies comparing PAG tests to a gold standard, routinely report sensitivity and specificity in addition to test agreement and NPV/PPV. Please incorporate this into the manuscript. 

Sensitivity and specificity have been added to Table 3.

Figures 1 and 2 do not, in this reviewer’s opinion, add additional insight into the data. While it shows a linear relationship of PAG OD to dpp for each parity group or dpp group, the following Figure 3 is a clearer depiction of the relationship of dpp OD, and assay cut off. It is this reviewer’s recommendation to remove Figures 1 and 2, keeping figure 3.

Figures 1 and 2 have been removed from the manuscript.

There are multiple mentions of the PAG levels “plateauing” during the postpartum period. Plateau typically indicates that something has reached its highest point and stays there but in this case, the PAGs have reached a low point. From a clarity standpoint for the reader, consider using different terminology such as “sustained nadir” or “baseline” to indicate the PAG levels have reached their lowest point and stayed there. 

This has been revised to sustained nadir. 

Lines 330-359 – There is extensive discussion of the previous studies and days at which the performance of the test was evaluated. Despite referencing Ricci, et al (Ref #10), early in the manuscript, there is no mention here of their finding that the IDEXX RPT has a period where PAG OD may drop into the recheck or inconclusive range (d 46-67 post AI) and was more pronounced in multiparous cows. Roberts et al 2015 also found this to be a gestational window where false negatives were possible in beef cows. This is of particular importance for beef producers using this test and understanding its limitations when the population they are testing may have a wide range of days of gestation depending on breeding strategy. These limitations identified by Ricci and discussed in both the Ricci and Roberts studies are hypothesized to be due to varying expression of PAG throughout early gestation. Given that the specific PAG in the lateral flow is unknown, is it possible that there could also be a window of gestation with this test that yields equivocal results or contributes to some of the false negatives depending on the expression pattern of the monoclonal PAG antibody used?

The authors agree with the reviewer that this is an important consideration; however, in the present study only 78 samples were collected from pregnant animals during that time point and of those samples all were above the 0.3 cutoff used to determine pregnancy. So this is something that cannot be evaluated with this data set.

Lines 378-384 – It is unclear what the process of uterine involution has to do with the use of a test to detect PAGs. Presumably, the authors are inferring that uterine involution is a critical step in return to fertility in the postpartum beef cow prior to the next conception. However, the way it is worded in this section is confusing. Please rework this paragraph to clarify the relationship you are making between uterine involution and the use of the blood-based pregnancy testing methods for the next pregnancy.

This sentence has been removed to prevent confusion. 

Reviewer #2: Specific Comments:

Title and Abstract

Appropriate title.

In abstract should probably be inexpensive rather than cheap? Line 27

This correction was made.

.. likelihood of false positive results are high if assays are performed fewer than 42 days pp. Might be less confusing (Lines 49-50.)

This correction was made.

Introduction

Line 72; only 22days necessary when describing as early as…

This correction was made.

Line 73: probably best to describe where the PAG’s are increasing- circulation?

This has been revised to read Pregnancy-associated glycoproteins steadily increase “in the maternal bloodstream” throughout gestation, are elevated at time of parturition, and then decrease after parturition [4,6,13].

Aims of study clearly stated.

M&M’s

IACUC stated- ethical research.

Study 1

It would be useful to know if those that were diagnosed pregnant via ultrasonography also calved, illustrating no pregnancy loss occurred to validate the PAG assay results. Otherwise the results would need to be assessed for the possibility of pregnancy loss and inherent PAG decreases.

That is not what this study was designed to evaluate. The objectives were to 1) validate the IDEXX Alertys OnFarm Pregnancy Test (lateral flow) and compare the accuracy of all three commercial PAG assays to transrectal ultrasonography and 2) to determine the postpartum interval necessary for clearance of pregnancy-associated glycoproteins from the previous pregnancy to avoid false positives. So, of the animals that were tracked to calving only 68 lost a pregnancy between first and final pregnancy diagnosis, and very few (<10) lost a pregnancy between final pregnancy diagnosis and calving. With these low numbers it is not possible to evaluate the data as a predictive test.

Study 2

Good array of time points.

Ultrasonography

Please state transducer frequency. Line 112

“5 MHz linear array probe” has been added to line 113. 

Please also state either conceptus was detected; or an embryo or fetus were detected. The embryo becomes a fetus at between 40-45 days depending on publications and interpretations. Similarly embryonic loss can only occur up to day 45; maybe best to describe as late embryonic or early fetal loss occurred? Lines 114-115.

This has been revised to read Pregnancy diagnosis was based on the visualization of “an embryo” or absence of one. A final pregnancy diagnosis occurred between d 30 and 80 following the end of the breeding season to determine if “early fetal” loss occurred. 

Lateral Flow

Please update the Figure descriptions, in line 127 the relevant figure is not numbered, and might not be featured in the SI?

There is an existing number, “1”, S1 Fig.

RVPT

The scoring for the RVPT appears to be a bit confusing, and should probably be a discussion point?

The authors Added the following: “, where a score of 0 had the same or less color than the negative control, a score of 1 had slightly more color than the negative control, a score of 2 had slightly less color than the positive control, and a score of 3 had the same or more color than the positive control. A numerical score of 0 or 1 would result in “no” meaning the female is not pregnant, while a numerical score of 2 or 3 would result in “yes” meaning the female is pregnant.”

Statistical analysis

Please use ultrasonography rather than ultrasound. It might be better to compare the tests in the same order as they are introduced: ultrasonography, lateral flow, RVPT and RPT, to make it easier for the reader.

Line 166

This has been revised in the materials and methods section, so it follows the order of the statistical analysis and results section.

My understanding is the ‘tendencies’ should not be part of statistical analysis.

The definition of what level was considered a tendency is stated in the statistical section.

Results

Table 1 useful.

For table 2 and for all others please alter ultrasound for ultrasonography.

Revised as suggested.

Please ensure all the figures are correctly documented in the article, and correlate with the relevant illustrated figures.

This has been revised 

Line 282; not sure that tendency should be recorded; maybe in the discussion. Similarly for line 290, and for all mentions of a tendency.

Tendencies were left in to allow readers to evaluate their own cutoff level, and the actual p-values are included with all statements.

Discussion and conclusion

Line 323; remove ‘a’ or member to singular.

Revised as suggested.

Lines 357-359: It is difficult to interpret the data without knowing the costs of the tests relative to the costs of ultrasonography. And, recognising the skill required to take the correct blood samples. If this could be addressed please.

Additional information has been added to address this concern. 

Lines 360-364:

Please discuss the relevance of a false positive compared with a false negative in relation to the managers requirements- feeding of a non-pregnant animal, or culling of a pregnant animal.

Additional information has been added to address this concern 

Discussion of relevance of DPP PAG results in relation to the likelihood of pregnancy if insemination is unlikely? And, importance of multiparity cf primiparity esp 35-49d pp. (Why?) (Lines 376-390)

This sentence has been removed

Line 388: Probably does not need: ‘with that said’ change made.; and to explain the differences RPT and Lateral Flow options more succinctly. I believe it already is. 

Revised as suggested. 

Line 408: A ‘shot’ although a common colloquialism, should probably be best described scientifically as some sort of drug administration/injection? 

Revised to injection. 

I think there needs to be some discussion as to the ‘immediacy’ of the gold standard ultrasonography compared with the relative delays associated with the tests described in the article. Particularly in relation to the handling and management of the animals in relation to the animals that are not pregnant.

Additional information has been added to this point

Some mention of the likelihood of a bovid in most management systems being inseminated within 50 days, hence the likelihood of being pregnant should be discussed in relation to the levels of PAG in post-partum cows. This in relation to the points mentioned in lines 442-446. (Uterine involution and voluntary waiting periods are typically more than 40 days in most dairy animals, and the effects of the described lactational anoestrus in beef animals typically resulting in breeding not occurring prior to 50days PP.)

Competing Interests

I think it is difficult to believe that no competing interests were declared when Idexx and Zoetis supported the study to a degree, and at least one author is an employee of Idexx.

This statement has been changed to list Dr. Jim Rhoades as an employee of IDEXX.

The Funding declaration in lines 452-453 appears to be at odds with that in the Financial disclosures section of the pre-manuscript section.

The financial statement has been added to and now reads. “This research did not receive any specific grant from funding agencies in the public, commercial, or not-for-profit sectors. This project was partially funded by donations from IDEXX and Zoetis.”

---

## [Decision Letter · Decision Letter 1]

24 May 2024

PONE-D-24-00715R1Evaluation of pregnancy associated glycoproteins assays for on farm determination of pregnancy status in beef cattlePLOS ONE

Dear Dr. Perry,

Thank you for submitting your manuscript to PLOS ONE. After careful consideration, we feel that it has merit but does not fully meet PLOS ONE’s publication criteria as it currently stands. Therefore, we invite you to submit a revised version of the manuscript that addresses the points raised during the review process.

There are a remaining few minor items that would need to be addressed - see comments from reviewer #1 below.

We look forward to receiving your revised manuscript.

Kind regards,

Angel Abuelo, DVM, MRes, MSc, PhD, DABVP (Dairy), DECBHM

Academic Editor

PLOS ONE

Journal Requirements:

Reviewers' comments:

Reviewer's Responses to Questions

**Comments to the Author**

1. If the authors have adequately addressed your comments raised in a previous round of review and you feel that this manuscript is now acceptable for publication, you may indicate that here to bypass the “Comments to the Author” section, enter your conflict of interest statement in the “Confidential to Editor” section, and submit your "Accept" recommendation.

Reviewer #1: (No Response)

Reviewer #2: All comments have been addressed

2. Is the manuscript technically sound, and do the data support the conclusions?

Reviewer #1: Yes

Reviewer #2: Yes

3. Has the statistical analysis been performed appropriately and rigorously? 

Reviewer #1: Yes

Reviewer #2: I Don't Know

4. Have the authors made all data underlying the findings in their manuscript fully available?

Reviewer #1: Yes

Reviewer #2: Yes

5. Is the manuscript presented in an intelligible fashion and written in standard English?

Reviewer #1: Yes

Reviewer #2: Yes

6. Review Comments to the Author

Reviewer #1: Thank you for the time and effort you have put into revising this manuscript. The revised version is much closer to completion and reads more easily than the first. However, there are a few additional minor revisions this reviewer would suggest prior to final acceptance.

Line numbers correspond to the marked up version of the manuscript

Line 63-65 This section, starting with “In order…” is a sentence fragment. Please revise so it is more clear and complete.

Line 237 – In table and in legend, please correct spelling of “sensitivity”

Lines 393-394 – In this sentence, it would be appropriate to use “ultrasound equipment” rather than “equipment of ultrasonography”

Lines 394-397 – A better way to discuss the costs would be to state that the blood test methods range from $4.50 to $8 per head while the cost for ultrasound is going to depend on the veterinarian’s rate to perform the test and their speed. Many veterinarians, working with an efficient crew, can run 30-45 cows through the chute in an hour (or more) likely making the cost of all methods relatively similar and maybe even less per head for ultrasonography since veterinarians typically charge an hourly rate for these services (rather than per head). It also needs to be mentioned that a skilled ultrasonographer can provide valuable information about the pregnancy (viability, stage of gestation, fetal sex, etc) that is not provided by a blood test. Because of the extreme variability in rates charged by practicing veterinarians providing ultrasonography services across the country, I would suggest not including that dollar figure. Additionally, the most logical argument for the use of a cow side test is in rural areas where producers are unable to find veterinarians to perform pregnancy diagnosis. This tool allows those producers to benefit from pregnancy diagnosis and identification of non-pregnant cows which will help their bottom line even when professional services are unavailable.

Lines 479-481 – The added sentences do not flow with the rest of the paragraph content. Please consider revising so they complete the paragraph. One suggested revision would be to word as follows: “Due to the additional time required for diagnosis of pregnancy with blood based tests, management decisions may be delayed compared to ultrasonography and may result in additional labor to sort open females once test results are available.”

Reviewer #2: Thank you for addressing the review comments; hopefully it has been beneficial to the publication to have done so.

I have two concerns, both of which I will leave to the editor/s to address:

I am not convinced that funding acquisition only is sufficient grounds for scientific authorship, although I am sure that J Rhodes did more than obtain funding, and would have been involved in the study concepts and editing the submitted document, thereby making an intellectual contribution to the research and it's output.

I think that 'tendency' should be left for the reader to determine, and not be suggested by the authors.

I am happy for the amended article to be published.

7. PLOS authors have the option to publish the peer review history of their article (what does this mean?). If published, this will include your full peer review and any attached files.

Reviewer #1: No

Reviewer #2: No

---

## [Author Response · Author response to Decision Letter 1]

13 Jun 2024

We greatly appreciate the effort you and the reviewers have made in reviewing this manuscript. We have incorporated almost all of the suggested comments, and our individual responses to each suggestion is listed below.

Line 63-65 This section, starting with “In order…” is a sentence fragment. Please revise so it is more clear and complete.

This has been revised to read “Pregnancy diagnosis within an operation is not only important, but necessary to increase profitability and have a complete and successful reproductive management program”

Line 237 – In table and in legend, please correct spelling of “sensitivity”

Revised as suggested

Lines 393-394 – In this sentence, it would be appropriate to use “ultrasound equipment” rather than “equipment of ultrasonography”

Revised as suggested

Lines 394-397 – A better way to discuss the costs would be to state that the blood test methods range from $4.50 to $8 per head while the cost for ultrasound is going to depend on the veterinarian’s rate to perform the test and their speed. Many veterinarians, working with an efficient crew, can run 30-45 cows through the chute in an hour (or more) likely making the cost of all methods relatively similar and maybe even less per head for ultrasonography since veterinarians typically charge an hourly rate for these services (rather than per head). It also needs to be mentioned that a skilled ultrasonographer can provide valuable information about the pregnancy (viability, stage of gestation, fetal sex, etc) that is not provided by a blood test. Because of the extreme variability in rates charged by practicing veterinarians providing ultrasonography services across the country, I would suggest not including that dollar figure. Additionally, the most logical argument for the use of a cow side test is in rural areas where producers are unable to find veterinarians to perform pregnancy diagnosis. This tool allows those producers to benefit from pregnancy diagnosis and identification of non-pregnant cows which will help their bottom line even when professional services are unavailable.

Revised as suggested

Lines 479-481 – The added sentences do not flow with the rest of the paragraph content. Please consider revising so they complete the paragraph. One suggested revision would be to word as follows: “Due to the additional time required for diagnosis of pregnancy with blood based tests, management decisions may be delayed compared to ultrasonography and may result in additional labor to sort open females once test results are available.”

This has been revised as suggested.

---

## [Editor Report · Decision Letter 2]

16 Jun 2024

Evaluation of pregnancy associated glycoproteins assays for on farm determination of pregnancy status in beef cattle

PONE-D-24-00715R2

Dear Dr. Perry,

We’re pleased to inform you that your manuscript has been judged scientifically suitable for publication and will be formally accepted for publication once it meets all outstanding technical requirements.

Kind regards,

Angel Abuelo, DVM, MRes, MSc, PhD, DABVP (Dairy), DECBHM

Academic Editor

PLOS ONE
---

## [Editor Report · Acceptance letter]

21 Jun 2024

PONE-D-24-00715R2 

PLOS ONE

Dear Dr. Perry, 

I'm pleased to inform you that your manuscript has been deemed suitable for publication in PLOS ONE. Congratulations! Your manuscript is now being handed over to our production team.

Kind regards, 

on behalf of

Dr. Angel Abuelo 

Academic Editor

PLOS ONE